# The Relationship between Concentration Effort, Focus Back Effort, Focus Back State, and Mind Wandering

**DOI:** 10.3390/bs14030162

**Published:** 2024-02-22

**Authors:** Hong He, Yunyun Chen, Xuemin Zhang

**Affiliations:** 1Institute of Brain and Psychological Sciences, Sichuan Normal University, Chengdu 610066, China; hehong@sicnu.edu.cn; 2Beijing Key Lab of Applied Experimental Psychology, Faculty of Psychology, Beijing Normal University, Beijing 100875, China; 202031061006@mail.bnu.edu.cn; 3State Key Laboratory of Cognitive Neuroscience and Learning &McGovern Institute for Brain Research, Beijing Normal University, Beijing 100875, China

**Keywords:** mind wandering, concentration effort, focus back effort, focus back state

## Abstract

Focus back effort, concentration effort, and focus back state are factors associated with mind wandering. Focus back effort, proposed in alignment with the definition of focus back state, has been previously regarded as one manifestation of concentration effort. The primary goals of the current study are to explore the relationship between concentration effort, focus back effort, focus back state, and mind wandering. To shed light on the issue, we assessed the level of each cognitive measure in a single task. The findings revealed significant correlations between concentration effort, focus back effort, focus back state, and mind wandering. Mediation analysis suggested that focus back effort played a mediating role in the relationship between concentration effort and focus back state. Additionally, we observed that these measures independently influenced task performance through their impact on mind wandering. Our results provide potential avenues for interventions aimed at addressing individuals’ mind wandering and enhancing task performance.

## 1. Introduction

Mind wandering is commonly described as thoughts that are unrelated to the current task [1], although the ongoing debate about its definition persists [2,3,4]. The prevalence of mind wandering has been substantiated through various studies [5,6,7,8], sparking considerable research interest in understanding its characteristics. In the present investigation, mind wandering was operationalized as thoughts not relevant to the ongoing activities. Researchers usually require individuals to do undemanding tasks where mind wandering can easily arise (e.g., sustained attention to response task, SART) to study this phenomenon [8,9].

There is a growing body of evidence indicating an association between mind wandering and concentration - in other words, attention, which has been proposed to be the opposite to what is found in the state of mind wandering [1]. There are several individual differences in the level of concentration. For instance, individuals experience more mind wandering and fewer on-task thoughts during activities that require less concentration [10]. The concept of concentration effort has been a focal point in mind wandering research, defined as the effort to concentrate on what an individual is doing [5,11]. Psychologists gauge concentration effort by asking individuals whether they are actively trying to concentrate on their current activity at the time of assessment, revealing that individuals exhibit decreased mind wandering frequency when trying to concentrate in daily life [5,12]. Kane et al. (2007) also demonstrated that mind wandering could be predicted by working memory capacity, specifically when activities demanded significant concentration. Moreover, they indicated that feelings of happiness and competence were found to increase the level of concentration effort. Additionally, the literature suggests that self-report ratings of concentration effort significantly negatively predict mind wandering [11,13].

Another relevant factor in the context of mind wandering is the focus back state. The focus back state was first developed by Hasenkamp et al. (2012), and has been explored using the focused attention (FA) meditation paradigm during functional magnetic resonance imaging (fMRI) scans. In the FA meditation task, participants were instructed to press a button when they detected themselves in a state of mind wandering and then refocus on the task [14]. Hasenkamp et al. (2012) proposed a cognitive fluctuation model in which the focus back state emerges before the sustained focus stage and after the awareness of mind wandering phase. In comparison to the mind wandering phase, focus back state was associated with activation in frontoparietal regions [14], belonging to the executive network [15,16]. Although there is no direct evidence demonstrating a significant correlation between the level of the focus back state and frequency of mind wandering, we suggest a negative correlation between these variables. It is reasonable to hypothesize that the higher the level of the focus back state experienced by an individual when they realize their minds have wandered, the fewer instances of mind wandering they would exhibit.

Recent findings have indicated that mind wandering is linked to a decreased level of focus back effort, defined as the effort to shift one’s attention from mind wandering to focusing on ongoing activities [17], that is, trying to experience the focus back state. Focus back effort is conceptualized based on the definition of the focus back state [17,18,19,20]. While focus back effort is considered a form of concentration effort [17,18], there are distinctions between the two measures. Specifically, concentration effort reflects an individual’s overall cognitive investment in a task, whereas focus back effort pertains to the degree of effort exerted in attempting to redirect attention during the process of mind wandering. In a prior study, mood induction procedures were employed to explore the causal relationship between mood and focus back effort, revealing that mood can influence focus back effort both in laboratory settings and in daily life [17]. Specifically, the positive mood induction group exhibited higher levels of focus back effort than the negative mood induction group. Subsequent work utilizing structural equation modeling (SEM) and resting-state fMRI demonstrates a connection between focus back effort and pleasantness of mind wandering [18]. Moreover, higher levels of focus back effort are predicted by increased motivation [20] and higher task demand [19], indicating that focus back effort can be influenced by domain-specific factors. We also suggested that variations in individual interests indirectly predict task performance, mediated by motivation, efforts to focus back, and mind wandering [19]. In other words, mind wandering could serve as a mediator in the relationship between focus back effort and task performance.

Given the prior statements or hypotheses suggesting correlations between concentration effort, focus back state, and focus back effort and considering that the definition of focus back effort was inspired by the focus back state, with focus back effort treated as a form of concentration effort in previous manuscripts [17,18], there may be a significant positive correlation between these variables. As a preview of our results, these measures were indeed found to be significantly positively correlated with each other. The subsequent goal of the investigation is to further scrutinize the relationship among these measures through a model. For instance, imagine you are reading a book, requiring a high level of concentration effort to stay focused on the material. During a mind wandering state, you may exert additional effort to focus back on the reading behavior, resulting in a higher level of focus back state. Consequently, we hypothesized that focus back effort may account for the relationship between concentration effort and focus back state. In comparison to participants reporting lower levels of concentration effort, those reporting higher levels of concentration effort are likely to exhibit higher levels of focus back effort, and this increased inclination toward focus back effort would positively predict the focus back state.

Mind wandering has been demonstrated to be accompanied by perceptual decoupling [21,22], a phenomenon suggested to be associated with impaired performance [23]. Extending this understanding, it can be inferred that, akin to how mind wandering mediates the relationships between focus back effort and task performance, mind wandering would also mediate the relationships between concentration effort and task performance, as well as between focus back state and task performance. In addition, we aimed to explore the potential mediation effect, examining whether concentration effort predicts focus back effort, with focus back effort subsequently predicting focus back state and the focus back state predicting mind wandering.

In sum, four key aspects were considered in the current study. First, we aimed to assess whether concentration effort, focus back effort, focus back state, and mind wandering are significantly correlated with each other. Second, our objective was to investigate whether focus back effort mediates the relationship between concentration effort and focus back state. Our third objective was to examine whether concentration effort predicts focus back effort, with the latter subsequently influencing focus back state, which in turn predicts mind wandering. The last aim was to explore whether, in a similar manner to focus back effort, concentration effort and focus back state can predict task performance by influencing mind wandering. In the present article, a SART was employed to investigate the issues of interest. While the meditation task is more suitable for assessing focus back state, it does not fully capture the real concentration effort and focus back effort of participants (as they are asked to refocus on the task when experiencing mind wandering states in the meditation task) [14]. Conversely, a SART with thought probes has been widely used when examining the properties of mind wandering and related measures [8,24]. Therefore, we considered the SART with thought probes as more appropriate for examining the properties of the three measures. The SART involves two types of errors: commission errors (failures to withhold responses to No Go trials) and omission errors (failures to respond to Go trials). Commission errors are indicated to be mainly driven by the speeding of reaction times, while omission errors are indicative of more pronounced mind wandering [9]. As outlined above, we employed omission errors as a behavioral indicator of task performance in the study because our aim was to explore the mediating effects concerning mind wandering-related task performance.

## 2. Methods

### 2.1. Participants

The sample consisted of 121 participants from universities in Beijing (38 males and 83 females; age range = 18–38 years, M = 21.81, SD = 2.95). The sample size was determined using the stopping rule, collecting data from as many individuals as possible before the conclusion of an academic term. No participants were excluded from the analyses. Each participant provided informed consent at the beginning of the experiment and received compensation for their participation. The study received approval from the Institutional Review Board of the Faculty of Psychology, Beijing Normal University.

### 2.2. Procedure

The procedure unfolded as follows: all participants provided informed consent and received instructions and practice before each formal experiment. Each participant engaged in two tasks, lasting approximately 50 minutes, administered in the following order: the attention network test [25] and the SART with thought probes. Subsequently, a set of questionnaires was administered to each participant. For the purposes of the present study, we focused solely on the results of the second task, the SART with thought probes, which comprised roughly 28 minutes of the session (the task is detailed in the Task section).

### 2.3. Task

#### 2.3.1. Sustained Attention to Response Task (SART)

As shown in Figure 1, the SART [26] was utilized in this study. The SART was programmed using the Psychtoolbox extension in Matlab, Version 2019a. Participants were tasked with responding to the digits 1–9 by pressing the space bar while refraining from responding to the infrequent target (number 3). Each digit (white) on a black background was presented for 1000 ms, followed by a black (blank) screen for 1000 ms. The experiment comprised four blocks of 126 trials, with each digit randomly presented 14 times within each block. Five probes were presented randomly, prompting participants to report their ongoing thoughts in each block. A total of 20 probes were administered. No target appeared for 5 seconds prior to each probe, and there were no instances where the number 3 immediately preceded or followed another number 3 or a probe. In the practice phase, participants completed a practice task consisting of 18 trials and 1 thought probe.

#### 2.3.2. Thought Probes

During the sustained attention to response task, participants were intermittently probed with thought probes, instructing them to report their ongoing thoughts during the 5 seconds preceding the probes. Prior to the experiment, participants received detailed instructions regarding the various thought probes. Specifically, participants were shown:(1)For the five seconds prior to being asked, was your attention on- or off-task? (1—completely on-task, 6—completely off-task)(2)To what extent were you trying to concentrate on the task? (1—not at all, 6—very much)(3)To what extent were you experiencing the focus back state? (1—not at all, 6—very much)(4)To what extent were you trying to focus back on the task? (1—not at all, 6—very much)

Each participant was required to assess their ongoing experiences on a six-point scale. A response of 1 on Question 1 was considered “on task”, while responses 2–6 on Question 1 were interpreted as indicating that participants had experienced mind wandering to some extent [27]. Questions 2–4 were presented independently of the answers to Question 1 to maintain consistency with prior research [8,28,29,30] and control for response bias. This approach ensured that individuals, having answered “on task”, were not influenced by subsequent questions and were less likely to choose this response simply to expedite the task, even if they did not genuinely experience on-task thoughts.

### 2.4. Statistical Analysis

Mind wandering was quantified by averaging the scores of probe-caught mind wandering. The focus back effort score was determined as the average score of ratings in Question 4 when the responses to Question 1 fell between 2 and 6. Simultaneously, the focus back state score was computed by averaging the ratings of Question 3, when the answers to Question 1 indicated that the participants’ minds had wandered. The concentration effort score (Question 2) was calculated by averaging all concentration effort estimates without factoring in the responses to Question 1.

We conducted a Pearson product moment correlation analysis using IBM SPSS-19. To test the hypotheses, mediation analyses with 5000 bootstrap samples and a 95% confidence interval (CI) level to assess the mediating effect were performed using the PROCESS modeling tool [31]. A significant indirect effect is acknowledged when the CI does not include 0, indicating that the mediating effect is different from 0 [32].

## 3. Results

Descriptive statistics for all primary measures of interest are presented in Table 1. As observed in the table, skewness and kurtosis values of omission errors exceeded acceptable ranges (skewness > 2, kurtosis > 4), indicating non-normal distributions for this measure. Consequently, we employed a square root transformation on these measures. This transformation successfully brought the skewness and kurtosis of omission errors into acceptable ranges (transformed skewness = 0.94, transformed kurtosis = 0.52). To clarify, the subsequent analyses utilized the transformed data of omission errors. There are no gender differences observed in these variables (*ps* > 0.05, Appendix A).

Subsequently, we examined the Pearson correlation coefficients for all primary measures. As depicted in Table 2, concentration effort, focus back effort, and focus back state were significantly positively correlated with each other: concentration effort × focus back effort (*r* = 0.76, *p* < 0.001), concentration effort × focus back state (*r* = 0.53, *p* < 0.001), and focus back effort × focus back state (*r* = 0.79, *p* < 0.001). Concentration effort and focus back effort were significantly negatively correlated with mind wandering (concentration effort × mind wandering: *r* = −0.48, *p* < 0.001, focus back effort × mind wandering: *r* = −0.43, *p* < 0.001). The results were consistent with the expected relation between focus back state and mind wandering (*r* = −0.25, *p* = 0.005). In alignment with prior findings [9,33], mind wandering was significantly positively correlated with omission errors (*r* = 0.30, *p* = 0.001). Partial correlations, denoting the correlations between variables of interest adjusted by regressing out age and gender, are presented in Appendix A. The findings were consistent with the correlations mentioned above.

In line with our hypothesis, the mediation results indicated that focus back effort mediated the relationship between concentration effort and focus back state (indirect effect = 0.61, bootstrapped 95% CI [0.452, 0.807]). This implies that focus back effort serves as a mechanism through which concentration effort influences the level of focus back state (see Figure 2A). The results of the chain mediation analysis revealed a significant direct path from concentration effort to mind wandering (*β* = −0.34, *p* = 0.008). The mediation process comprised three indirect effects: path 1 (concentration effort → focus back effort → mind wandering, indirect effect = −0.146), path 2 (concentration effort → focus back state → mind wandering, indirect effect = −0.026), and path 3 (concentration effort → focus back effort → focus back state → mind wandering, indirect effect = 0.119). The bootstrap 95% confidence intervals for these paths included 0 (path 1: −0.400, 0.096; path 2: −0.119, 0.009; path 3: −0.056, 0.338), suggesting that these paths were not statistically significant (see Figure 2B).

Mediation models revealed nonsignificant direct paths from concentration effort to omission errors (Figure 3A), from focus back effort to omission errors (Figure 3B), and from focus back state to omission errors (Figure 3C), respectively. Furthermore, mind wandering mediated the relationships between concentration effort and omission errors (indirect effect = −0.01, bootstrapped 95% CI [−0.300, −0.060], Figure 3A), between focus back effort and omission errors (indirect effect = −0.01, bootstrapped 95% CI [−0.024, −0.005], Figure 3B), as well as between focus back state and omission errors (indirect effect = −0.01, bootstrapped 95% CI [−0.017, −0.002], Figure 3C).

## 4. Discussion

This study aimed to address questions regarding the relationship between concentration effort, focus back effort, focus back state, and mind wandering. The findings revealed significant correlations between concentration effort, focus back effort, and focus back state. Additionally, we extended these results by establishing focus back effort as a mediator in the relationship between concentration effort and focus back state. However, the sequential mediating effect of concentration effort → focus back effort → focus back state → mind wandering was found to be non-significant. Nevertheless, we demonstrated that all three measures independently influence task performance through their impact on mind wandering. The following explanation delves into the significance of considering concentration effort, focus back effort, and focus back state in the study of mind wandering.

The correlation results revealed significant positive associations among concentration effort, focus back effort, and focus back state. These findings align with the proposition that focus back effort, defined based on the concept of focus back state, is considered one form of concentration effort [17,18]. In line with prior research, concentration effort [11] and focus back effort [17] were significantly correlated with mind wandering. Additionally, focus back effort exhibited a significant negative correlation with mind wandering, indicating that individuals with higher levels of focus back state experience less mind wandering. These outcomes suggest a potential role for these measures in mitigating mind wandering. Moreover, no significant relationships were observed between each of the three measures and omission errors, implying that these measures do not directly impact task performance. The negative relationship between mind wandering and task performance in the current study is consistent with existing evidence demonstrating the detrimental impact of mind wandering across various tasks, such as response inhibition [34], reading comprehension [35], and driving [36].

Moreover, focus back effort was validated as a mediator in the relationship between concentration effort and focus back state. Consequently, focus back effort may contribute to the observed tendency for individuals with higher levels of concentration effort to experience elevated focus back states. The result suggests that interventions meant to increase focus back state should take the variability of concentration effort and focus back effort into account. Previous research has often framed concentration effort as a cognitive response to task-related factors, including task significance and difficulty [5,11]. Concentration effort and motivation are both domain-specific factors susceptible to changes in the environment. The predictive influence of concentration effort on focus back effort aligns with prior research highlighting the substantial impact of task difficulty and motivation on focus back effort [19]. Focus back effort serves as an indicator of executive control regulation during task processes [18,19,20]. Consequently, these findings support the role of domain-specific factors, specifically how concentration effort modulates executive control during tasks. Furthermore, the results of this mediating model extend previous research by demonstrating that focus back effort is not only linked to neural characteristics associated with focus back state [18] but can also predict focus back state.

However, when we integrated mind wandering into the mediating model, the path concentration effort → focus back effort → focus back state → mind wandering became non-significant. Subsequent findings indicated that concentration effort, focus back effort, and focus back state contributed to task performance by independently reducing omission errors through the mitigation of mind wandering. Although none of the three measures directly impacted task performance, they individually influenced task performance by diminishing mind wandering. These results lend support to the positive roles of concentration effort, focus back effort, and focus back state in relation to task performance. This opens up potential avenues for targeted interventions to enhance concentration effort, focus back effort, and focus back state, thereby reducing the frequency of mind wandering and improving task performance. Given that mind wandering has been associated with impaired performance across a range of tasks [34,35,36,37], this study carries significant implications for performance in diverse contexts. Moreover, these mediation results suggest that mind wandering is a process that can be subjectively controlled. These findings provide novel perspectives for subsequent studies aiming to intervene in mind wandering and task performance.

While we consider the findings of this study to provide novel insights into the connections among concentration effort, focus back effort, focus back state, and mind wandering, several limitations and future directions merit consideration. One notable limitation is that we exclusively assessed these measures of interest in a laboratory setting; thus, the generalizability of the results to daily life requires verification. Another point of consideration is that we did not employ the same methodology as previous researchers in measuring focus back state. There were differences between the focus back state we measured and that assessed in previous studies, where participants were instructed to refocus on the task, and focus back state occurred after the process of meta-awareness of mind wandering [14], whereas we collected self-reported focus back state without the guidance of refocusing in the current study. Caution is needed when interpreting the results of the current study in light of these differences. Additionally, the study did not account for current concerns, which occupy a substantial portion of individuals’ mind wandering thoughts [38], when exploring the mediation effects among these measures, mind wandering, and task performance. Future studies will aim to address these concerns of mind wandering. Moreover, given the proven significance of motivation and interest in predicting focus back effort, future research should integrate motivation and interest into the interconnected study of concentration effort, focus back effort, focus back state, and mind wandering. Mind wandering takes on diverse forms, encompassing both aware and unaware states, as well as intentional and unintentional occurrences [24,39,40]. Additionally, the emergence of the focus back state occurs after the awareness of the mind wandering phase [14]. Subsequent research is warranted to explore the relationships among these factors and these dimensions of mind wandering. Moreover, causal interpretations will be possible in future studies with intervention experiments. Specifically, interventions targeting concentration effort, focus back effort, and focus back state have the potential to effectively reduce the occurrence of mind wandering. Accordingly, in work settings, interventions focused on enhancing these factors may lead to improved job performance. Similarly, within educational contexts, the enhancement of these factors could help mitigate the adverse impact of mind wandering on academic performance. We did not collect information on the education/occupation of the participants. Subsequent studies could further explore the correlation between education/occupation and concentration effort, focus back effort, focus back state, and mind wandering.

## 5. Conclusions

Taken together, the results suggest that concentration effort, focus back effort, focus back state, and mind wandering correlated with each other. Furthermore, focus back effort serves as a mediator in the relationship between concentration effort and focus back state. Although the chain-mediating effect becomes non-significant when accounting for mind wandering, these factors independently contribute to predicting task performance by influencing mind wandering. While the current study may not provide complete clarity in the field, the results of the present study increase our understanding of concentration effort, focus back effort, focus back state, and their internal correlations. We encourage researchers to consider the crucial roles of concentration effort, focus back effort, and focus back state in their studies of mind wandering.

## Figures and Tables

**Figure 1 behavsci-14-00162-f001:**
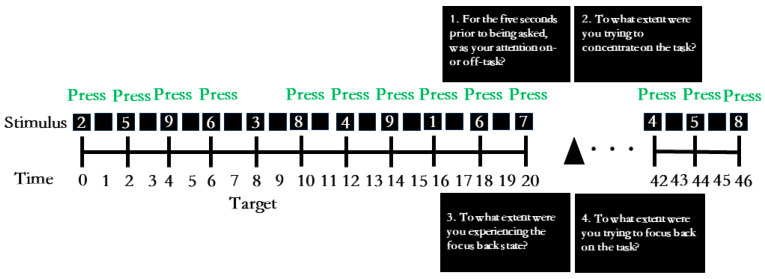
The sustained attention to response task.

**Figure 2 behavsci-14-00162-f002:**
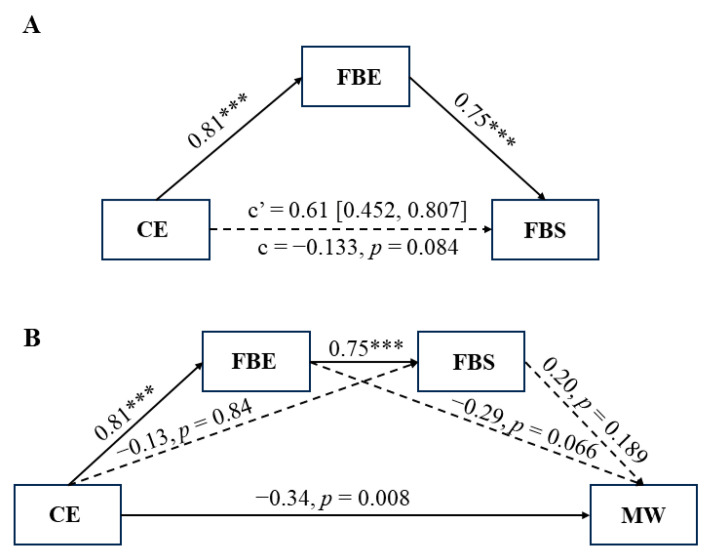
(**A**) Mediation models illustrating the relationship between concentration effort (CE) and focus back state (FBS), with focus back effort (FBE) serving as a mediator. (**B**) Mediation effect paths of focus back effort and focus back state between concentration effort and mind wandering (MW). The coefficients presented in this figure are in their unstandardized form. Continuous lines denote significant relationships at *p* < 0.05, while dashed lines indicate non-significant relationships at *p* < 0.05. *** *p* < 0.001.

**Figure 3 behavsci-14-00162-f003:**
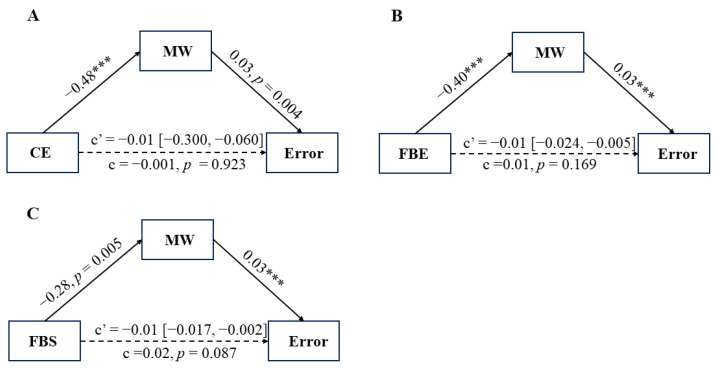
Mediation models illustrating (**A**) the relationship between concentration effort and omission errors with mind wandering as a mediator, (**B**) the relationship between focus back effort and omission errors with mind wandering as a mediator, and (**C**) the relationship between focus back state and omission errors with mind wandering as a mediator. The indirect effect is reported as c’, and the direct effect is presented as c. The coefficients presented in this figure are in their unstandardized form. CE = concentration effort; FBE = focus back effort; FBS = focus back state; MW = mind wandering. Continuous lines represent significant relationships at *p* < 0.05, whereas dashed lines indicate non-significant relationships at *p* < 0.05. *** *p* < 0.001.

**Table 1 behavsci-14-00162-t001:** Descriptive statistics for all primary variables.

Measure	Mean (SD)	Range	Skewness	Kurtosis
CE	3.88 (0.86)	1.45–5.75	−0.31	−0.04
FBE	3.50 (0.92)	1.26–5.50	−0.17	−0.40
FBS	3.28 (0.77)	1.60–5.00	−0.06	−0.50
MW	3.04 (0.86)	1.20–5.35	0.36	−0.22
Omission errors	0.01 (0.02)	0–0.11	2.61	8.10

Note. CE = concentration effort; FBE = focus back effort; FBS = focus back state; MW = mind wandering.

**Table 2 behavsci-14-00162-t002:** Correlation coefficients for all primary variables of interest.

Measure	CE	FBE	FBS	MW	Omission Errors
CE	-	0.76 ***	0.53 ***	−0.48 ***	−0.15
FBE		-	0.79 ***	−0.43 ***	−0.02
FBS			-	−0.25 **	0.07
MW				-	0.30 **
Omission errors					-

Note. CE = concentration effort; FBE = focus back effort; FBS = focus back state; MW = mind wandering. *** *p* < 0.001, ** *p* < 0.01.

## Data Availability

The datasets generated and/or analyzed during the current study are available from the corresponding author upon reasonable request. These studies were not preregistered.

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
