# Peer review of "The Relationship between Concentration Effort, Focus Back Effort, Focus Back State, and Mind Wandering"

_behavsci, 2024, doi:10.3390/bs14030162_

Round 1

Reviewer 1 Report

Comments and Suggestions for Authors

Dear authors,

thank you for your manuscript. The relationship between concentration, focus back effort and state and mind wandering was very clear and structured described.

I have one comment: You write that the higher the level of focus back state the lower the frequency of mind wandering. But could it not be that this affects the length rather than the frequency of mind wandering?

I further detected one spelling error: line 243 there is one zero too much in the CI.

Best regards

Reviewer 2 Report

Comments and Suggestions for Authors

It's interesting to know the correlation and association between the domains of concentration effort, focus back effort, focus back state and mind wandering.  The authors have clearly demonstrated their objectives of the study and the model to predict the correlation between different traits. 

1. Did the authors acquire the education/occupation of the participants.  If so, have they observed any correlation between education/occupation with respect to concentration effort, focus Back Effort, focus back state and mind wandering. 

2. Likewise, any difference or correlation is noticed between male to female subjects.

3. suggestible to add a picture showing the performance of the task (SART).

Reviewer 3 Report

Comments and Suggestions for Authors

The authors explored the relationship among concentration effort, focus back effort, focus back state, and mind wandering using a single task. The findings revealed significant correlations among these four aspects and focus back effort played a mediating role in the relationship between concentration effort and focus back state. The manuscript is well-written and the topic is interesting. I have only minor concerns.

1. How many probes of questions are there in total throughout the 28-minute task? Is the number of measures sufficient and reliable to reflect the level of different factors investigated?

2. Did the author try using partial correlation? That is by controlling the influence of other factors? Will the result change?

3. Can the authors have some real examples for the suggestions of intervention in the discussion?
